# Etorphine-Azaperone Immobilisation for Translocation of Free-Ranging Masai Giraffes (*Giraffa Camelopardalis Tippelskirchi*): A Pilot Study

**DOI:** 10.3390/ani10020322

**Published:** 2020-02-18

**Authors:** Francesca Vitali, Edward K. Kariuki, Domnic Mijele, Titus Kaitho, Massimo Faustini, Richard Preziosi, Francis Gakuya, Giuliano Ravasio

**Affiliations:** 1Dipartimento di Medicina Veterinaria, Università degli Studi di Milano, Via dell’Università 6, 26900 Lodi, Italy; massimofaustini@gmail.com (M.F.); giuliano.ravasio@unimi.it (G.R.); 2Department of Veterinary Services, Kenya Wildlife Service, P.O. Box 40241-00100 Nairobi, Kenya; ekariuki@kws.go.ke (E.K.K.); dmijele@kws.go.ke (D.M.); titus@kws.go.ke (T.K.); gakuya@kws.go.ke (F.G.); 3Ecology and Environment Research Centre, Department of Natural Sciences, Faculty of Science and Engineering, Manchester Metropolitan University, Manchester M1 5GD, UK; r.preziosi@mmu.ac.uk

**Keywords:** Masai giraffe, giraffe capture, chemical immobilisation, translocation, etorphine, azaperone, naltrexone, capture stress, giraffe physiology, blood gas analysis

## Abstract

**Simple Summary:**

Due to their peculiar anatomy and sensitivity to drugs, giraffes are among the most challenging mammals to immobilise. Masai giraffes have recently been listed as endangered. Hence, their conservation needs actions that require veterinary capture such as translocations. In this study, we evaluated a new protocol of immobilisation for translocation of free-ranging Masai giraffes. The hypothesis is that, by combining a potent opioid with a tranquiliser, it is possible to mitigate the capture stress, which is a major cause of disastrous homeostatic consequences, including capture myopathy and death. The combination produced, in all individuals, smooth and quick inductions and reliable immobilisations. Although hypoxaemia in a few individuals and acidosis were seen, the overall cardiorespiratory function was adequate. Whereas the initial stress to the capture was limited in the individuals, likely due to tourism-related habituation, the opioid-related excitement and resulting increased exertion was responsible for worse immobilisation and physiological derangement. A low dose of an antagonist was used and evaluated and, in the two-week boma follow-up, it proved to be efficient in providing safe recoveries and transport. At the investigated doses, the combination provided partially reversed immobilisation that allowed uneventful translocation in free-ranging Masai giraffes.

**Abstract:**

Etorphine-azaperone immobilisation was evaluated for translocation of Masai giraffes. Nine giraffes were darted with 0.012 ± 0.001 mg/kg etorphine and 0.07 ± 0.01 mg/kg azaperone. Once ataxic, giraffes were roped for recumbency and restrained manually. Naltrexone (3 mg/mg etorphine) was immediately given intravenously to reverse etorphine-related side effects. Protocol evaluation included physiological monitoring, blood-gas analyses, anaesthetic times, and quality scores (1 = excellent, 4 = poor). Sedation onset and recumbency were achieved in 2.6 ± 0.8 and 5.6 ± 1.4 min. Cardio-respiratory function (HR = 70 ± 16, RR = 32 ± 8, MAP = 132 ± 16) and temperature (37.8 ± 0.5) were stable. Arterial gas analysis showed hypoxaemia in some individuals (PaO_2_ = 67 ± 8 mmHg) and metabolic acidosis (pH = 7.23 ± 0.05, PaCO_2_ = 34 ± 4 mmHg, HCO_3_^−^ = 12.9 ± 1.2 mmol/l). Minor startle response occurred, while higher induction-induced excitement correlated to longer inductions, worse restraint, and decreased HCO_3_^−^. After 19 ± 3.5 min of restraint, giraffes were allowed to stand and were loaded onto a chariot. Immobilisations were good and scored 2 (1–3). Inductions and recoveries were smooth and scored 1 (1–2). Translocations were uneventful and no complications occurred in 14-days boma follow-up.

## 1. Introduction

Giraffe populations are declining and they are undergoing a silent extinction. The Masai giraffe (*Giraffa camelopardalis spp. tippelskirchi*), which is found only in East Africa, might be listed soon as a separate species as debated in recent genetic studies [1,2,3,4]. Masai giraffes have been declared endangered mainly due to poaching and habitat loss, and their status calls for urgent conservation actions [5].

Translocations are increasingly important tools for endangered species conservation to increase genetic exchange between isolated populations or to assist the recovery of declining populations [6]. Translocating wild giraffes is challenging mainly due to the potential dangers posed by their size, the harsh environmental conditions, and the lack of proper anaesthetic equipment in the field [7,8]. Chemical capture is usually needed in order to attach the ropes necessary to lead the giraffes onto a chariot (i.e., a trailer modified to transport giraffes). The common procedure for long distance translocations requires a boma confinement for two to three weeks before a long journey is faced [9].

Giraffe captures have been defined as the “art and science of giraffe immobilisation” due to the numerous challenges faced because of their unique anatomy and physiology [10]. Dramatic consequences are not rare and can result in unacceptable morbidity and mortality [7,8,10]. Anatomical features of giraffes make their handling very complicated. They are heavy and are prone to develop stress and injure their long neck. Additionally, aggressive movement of their legs can pose a danger to the capture team [7,8,10,11]. Giraffes are among the most susceptible African ungulate species to capture myopathy [12]. Because of their unique cardiovascular physiology, the giraffe heart may easily get damaged from oxygen debt, complicated by their smaller lung volumes and low compliance [13]. Their delicate physiological balance can easily be altered by potent opioids used for captures because giraffes are particularly sensitive and are prone to side effects [8]. Opioid-related respiratory depression and alterations in the thermoregulatory response commonly occur and worsen the oxygen debt and the acid-base derangement, and, in case of stress-driven hyperthermia, further increase the metabolic demand [8]. Furthermore, not only the immense physical activity, but also the psychological stress developed during the capture event produce a severe homeostatic imbalance that can lead to capture myopathy and death [12,14,15]. The high morbidity and mortality rates (sometimes >10%) encountered with previously reported opioid-based protocols has resulted in a hesitancy to anaesthetise this species [16]. In the field, to keep induction times and physical exertion to a minimum, giraffes are usually knocked down by overdosing with a potent opioid, which is immediately reversed when the giraffe is recumbent. This is followed by manual restraint. Naltrexone, which is a long-acting pure opioid antagonist that is given at dose rates between 5–100 mg per mg of etorphine, is often used, and is able to fully antagonise etorphine action and side effects, even though there are some controversial opinions on the effective doses [7,8,17]. This anaesthetic technique greatly limits what can safely be done to giraffes while they are recumbent, as they are completely awake [7,8,9,10].

Azaperone is a butyrophenone tranquiliser that has been widely used as an adjunct to an opioid-induced chemical restraint. Azaperone has sedative, anti-anxiety, and mild muscle relaxant effects, and, due to its alpha1 antagonistic effect, induces vasodilatation that can counterbalance opioid hypertension [18,19,20,21]. Reports of adverse effects in wild ungulates, such as severe hypotension and extrapyramidal effects, are rare and thought to result from high intravenous doses in stressed animals [19,22]. Guideline advice is that giraffes should be fully reversed before loading and that neuroleptics should not be given while being transported due to the danger of disorientation, unsteadiness on their feet, and collapse [9]. However, from previous experiences of the authors, it emerged that azaperone at low doses might provide useful tranquilisation during the physical restraint phase, when the giraffes are recumbent and reversed, which decreases the insurgence of manipulation stress without affecting the physiological function. Having calmer individuals might also benefit the safety of the loading and transport operations for both the animals and the capture team. The aim of this preliminary study was to investigate the physiological and handling safety of etorphine combined with azaperone for the immobilisation and translocation of free-ranging Masai giraffes. The evaluation of low doses of naltrexone in its efficacy for etorphine antagonisation was also performed by taking advantage of a two-weeks boma confinement follow-up.

We assumed that both the initial stress associated with the darting procedure and the physical exertion resulting from a mix of opioid-related excitement and anxiety influence the insurgence of acid-base and cardio-respiratory function derangements and their severe complications. Since they have different pathways and different prevention strategies, we aimed to investigate their role in affecting the physiological imbalance and the immobilisation outcome.

## 2. Materials and Methods

### 2.1. Animals, Drugs, and Procedures

This study was performed in the Rift Valley Region in Kenya, in the area surrounding Lake Naivasha, which is a forest region situated at 1884 m of altitude. It was performed on giraffes that were part of a translocation carried out for management and conservation purposes, organised by the Kenya Wildlife Service, which is the authority for wildlife and parks management in the country. The project required the capture of free-ranging individuals of Masai giraffe from the Naivasha area, in Nakuru County, and the subsequent translocation to a wildlife sanctuary in the Mombasa County.

The Kenya Wildlife Service (KWS) Department of Veterinary Services and the Biodiversity Research and Monitoring Office (KWS/BRM/5001) approved the project, which complied with the KWS guidelines to conduct research on wild mammalian species.

Nine free-ranging Masai giraffes, five females and four males, both subadults and young adults were considered for this study. The animals were captured over three consecutive mornings during which the environmental temperature was always between 18 and 25 °C. The capture was planned during the dry season in order to have a favourable terrain, even though the area is characterised by spots with thick vegetation and rough terrain. This requires particular care during the induction phase. Individuals suitable for the translocation were first spotted from a vehicle, and, after estimating the age and body size, and checking their apparent health status, a 3-mL dart (Dan-Inject 3 mL, 2.0 × 60 mm needle, S300 Syringe Dart, Dan-inject International, Skukuza, South Africa) containing the drug mixture was prepared. The immobilisation protocol included a combination of 8 or 9 mg of Etorphine (Captivon 9.8 mg/mL, Wildlife Pharmaceuticals, White River, South Africa) and 50 mg of Azaperone (100 mg/mL, Kyron Laboratories, Johannesburg, South Africa) depending on the estimated size of the giraffe. The individuals were cautiously approached in a vehicle, and when a proper distance was reached (15–20 m), a CO_2_ pressurised dart gun (Model JM; Dan-inject International, Skukuza, South Africa) was used to deliver the dart syringe intramuscularly in the upper hind leg or shoulder area. After being darted, the individuals were observed from a distance until the first signs of sedation were observed. When the giraffes appeared severely ataxic, and, in order to facilitate their recumbency and prevent them from both overexertion and falling in undesired areas, they were casted with ropes from an experienced capture team. A heavy rope, at chest height, was placed in the path of the giraffes, and, as the giraffes encountered the rope, the ends were crossed behind the hindquarters, which stopped the forward motion of the giraffes and led to recumbency [11]. Once the individuals were recumbent, naltrexone (3 mg/mg etorphine, 40 mg/mL Kyron Laboratories, Johannesburg, South Africa) was administered intravenously to reverse etorphine, in order to antagonise the opioid-related side effects. After the reversal, in order to maintain the lateral recumbency, the giraffes were manually restrained by holding the neck against the ground, and a blindfold was applied to minimise stress.

During the recumbency phase, the capture team positioned the ropes needed to subsequently load the individuals onto the chariot. During this time, blood and tissue samples were collected, age was estimated, and body measurements were recorded. The body weight of each giraffe was estimated from the length and girth measurements following the method described by Hall-Martin [23]. When the giraffe was ready to be loaded, the neck restraint was lifted, and the giraffe was allowed to stand up. Once standing, the blindfolded giraffe was guided through the use of the ropes, onto the chariot, and then transported to a boma 5–10 km away. The nine giraffes were housed in the boma for 14 days after which they have been translocated to a new site with a hard release [9].

### 2.2. Monitoring

Anaesthetic times and appositely descriptive scores were used in order to standardise the evaluation of the immobilisation procedure quality. The time from the darting (referred as time 0) to the first signs of sedation (“sedation time”), the successful application of ropes (“roping time”), when recumbency was reached (“recumbency time”), were recorded. The distance that the giraffes walked while chased by the darting vehicle before they were successfully darted, and the distance from the darting spot to the recumbency site were estimated.

*Ad-hoc* qualitative scores were created and used for each individual in order to differentiate between the initial behavioural reaction of the giraffe to the stress of the darting procedure, the “startle response”, from the excitatory behaviour displayed after the drugs showed the first effects, the “induction induced excitement”, which is characterised by intense physical activity and is likely due to both opioid-induced central excitation and anxiety.

The “startle response score” (Table 1) took into account the individual’s reaction to the darting vehicle, and the resulting duration and velocity of the chasing phase.

Table 2 referred to the individual’s behaviour after the first signs of sedation occurred. It was created to describe solely the degree of excitement and the consequent physical exertion displayed by the giraffes, which results from the excitatory effects of the opioid during the induction phase. Although this score mainly focused on the physical component of the capture stress, it is not possible to exclude that the excitatory behaviour is not only influenced by the drugs, but also by the anxious response of the giraffes to the perception of sedation signs. The behaviours considered for the excitement score were star gazing, high gait stepping, and ataxic accelerated ambling or galloping with no regard to the surrounding environment.

In order to evaluate the quality of induction, a descriptive score was used to define the safety of rope-assisted recumbency for the giraffes and the capture team (Table 3).

Once giraffes were recumbent, the time of administration of naltrexone was recorded. The physiological function and behaviour of the giraffes were monitored continuously throughout the recumbency. Since etorphine was early antagonised, the recumbency was maintained thanks to a combination of manual restraint and tranquilisation with azaperone. Occurrence of regurgitation and of other complications was recorded. The first record (T1) was made 2 min after naltrexone was administered IV, and the following records were made every 5 min (T2, T3).

The respiratory rate (RR) was monitored through chest movement observation and heart rate (HR) by auscultation of the heart with a stethoscope. Body temperature (T) was measured with a digital thermometer inserted in the rectum (Veterinary rectal thermometer, 25588 Gima S.p.a., Gessate, Italy). A pulse oximeter with a transmission probe (MD 300 Handheld Pulse Oximeter, ChoiceMMed Co., Tianjin, China) was attached to the rectal or vulvar mucosa to measure haemoglobin oxygen saturation (SpO_2_). Non-invasive blood pressure (NIBP) was measured oscillometrically using a cuff placed on the tail (Omron Digital Blood Pressure Monitor Model HEM-400 C, Omron Corporation, Kyoto, Japan).

At time T2, arterial blood was collected anaerobically from an ear artery by using a 1-mL heparinised syringe and analysed within 15 min using a portable blood gas analyser (VetStat Electrolyte and Blood Gas Analyser and Respiratory/Blood Gases cassettes, IDEXX Laboratories Italia Srl, Milano, Italy). The analysis included measured values for pH, partial pressure of arterial carbon dioxide (PaCO_2_), partial pressure of arterial oxygen (PaO_2_), sodium (Na), potassium (K), and chloride (Cl). Base excess (BE), bicarbonate (HCO_3_^−^), anion gap (AG), arterial haemoglobin oxygen saturation (SaO_2_), and haemoglobin (Hb) were calculated by the blood gas analyser from measured variables. The alveolar-to-arterial oxygen tension gradient [P(A-a)O_2_] was calculated by subtracting PaO2 measured by the blood gas analysis (BGA) from the alveolar oxygen tension (PAO_2_). PAO_2_ was calculated with the following equation:$$PAO2=FiO2\left(Pb-PH2O\right)-\frac{PaCO2}{RQ}$$
where FiO_2_ is the fraction of inspired oxygen (21% for room air), Pb is the barometric pressure measured during the study by the blood gas analyser, PH_2_O is the partial pressure of vapor in the alveoli (47 mmHg), PaCO_2_ is the partial pressure of carbon dioxide measured by the BGA, and RQ is the respiratory quotient, dependent on metabolic activity and diet, and is considered to be 1.0 for ruminants [24]. Since the capture site was not at sea level, we calculated the PaO2 expected for the average altitude we worked at (i.e., 1884 m) in order to use it as the cut-off value for defining hypoxaemia, using the alveolar-to-arterial oxygen tension formula. Assuming a normal alveolar-oxygen tension difference of 15 mmHg, the expected PaO_2_ was calculated by subtracting 15 mmHg from the calculated PAO_2_ value (as described above, but assuming a barometric pressure of 604 mmHg and PaCO_2_ of 35 mmHg for this estimation, which is considered the physiological reference value at this altitude) [25,26]. The resulting expected PaO_2_ at 1884 m of altitude was 66 mmHg, and animals with PaO_2_ values lower than this were considered hypoxaemic.

Restraint quality was assessed using an ad hoc descriptive qualitative score (Table 4), which considered the reactive behaviour of the individuals to human manipulation and painful stimuli like arterial punctures.

The total time of the duration of the recumbency phase (“restraint length”), the time that occurred from when the manual restraint was lifted to standing (“standing time”), the time that occurred for loading the giraffes on the chariot (“loading time”), and the duration of the transport to the boma were recorded. Descriptive scores were used to define the quality and safety of the recovery (“recovery score”) and of the loading procedures (“loading score”) (Table 5 and Table 6).

Giraffes’ behaviour was observed during the chariot transport from the capture site to the boma and adverse reactions were recorded. During the boma confinement, giraffes were monitored in order to assess their health status, welfare, and possible signs of re-narcotisation and capture myopathy. Giraffes’ apparent health status was recorded every hour for the first 12 h, and at least every 6 h for the remaining 14 days. Personnel from the capture team were always available on site and were instructed to record the occurrence of possible complications.

### 2.3. Data Analysis

Statistical analyses were performed using JMP, version 7.0 for Windows (SAS Institute, Cary, United States). Numeric data are presented as mean values ± standard deviation, with ranges where relevant. Scores are presented as median values with ranges. A non-parametric Spearman correlation coefficient was used to evaluate the correlations between the scores and the measured variables. The correlation coefficients and significances have been calculated for all pairs of variables. Only the most relevant statistical results are reported. A Student’s t-test was used to evaluate the difference between SpO_2_ and SaO_2_ values. The statistical method was chosen on the basis of the nature of data (mixed measured/score data) and of the sample size. A *p*-Value below 0.05 was considered significant.

## 3. Results

Data collected from all nine Masai giraffes were included in this study. In all the individuals, etorphine-azaperone combination provided a reliable immobilisation and no additional doses were needed. According to estimations made during restraint, the individuals weighed 722 ± 97 kg and were 3.6 ± 0.7 years old. Administered doses of etorphine were 0.012 ± 0.001 mg/kg (range 0.011–0.013 mg/kg) and azaperone 0.07 ± 0.01 mg/kg (range 0.063–0.083 mg/kg). Etorphine and azaperone doses in mg/kg administered to individuals of different sizes showed little variations, and there was no correlation between the variations in the doses in mg/kg and times for the first signs of sedation and casting nor induction time and score. Naltrexone was given intravenously at 0.036 ± 0.003 mg/kg (range 0.034–0.04 mg/kg), which reflects a 3:1 ratio with an etorphine dose. No dart failures were experienced and no individuals required a second dart. The dart impact sites were the muscles of the shoulder or the hindquarters, and all were considered excellent sites. The terrain was dry for all nine immobilisations, but, in six immobilisations, the area was characterised by a thick vegetation and rough terrain that required particular care during the induction.

The median startle response score was 2 (range 1–3) and showed some variability. Three giraffes appeared completely undisturbed by the presence of the darting vehicle. Four giraffes calmly walked away only when the vehicle was closer than 10 m. Two giraffes seemed to be uncomfortable and kept a distance from the vehicle. However, they always kept a relaxed gait and never broke into a gallop in this phase. Most giraffes were darted while they were eating and mostly within the herd. After being darted, they stayed on the spot or moved less than 20 m. All the giraffes quickly became calm and returned to the previous activity until the first signs of sedation occurred. The first signs of initial drug effects were observed within 2.6 ± 0.8 min (range 1.5–3.4 min) and included signs of ataractic tranquilisation defined as a decreased interest for external stimuli, increased motor activity with an ataxic gait, and isolation from the herd. In all giraffes, an excitatory phase was observed 1–3 min after darting, which was characterised by a sudden gallop or accelerated amble usually following the rising of ataraxia. The induction-induced excitement score also showed high variability among individuals. The median value was 2 and ranged between 1 and 4. Most of the giraffes showed a mild excitement. In three giraffes, the excitement lasted less than one minute until they were successfully roped down and was characterised by a low velocity amble, and three other giraffes displayed a mixed speedy amble/slow gallop for a couple of minutes. In three giraffes, a severe excitement was observed: a high velocity gallop was observed and maintained for up to 4 min and, in one case, roping was considered dangerous. According to the Spearman correlation test, the startle response developed during the darting procedure and the successive opioid-related excitement were not correlated. The startle response score also did not show correlations with any other score, anaesthetic time, or physiological variable. A higher induction-induced excitement score (i.e., greater excitement) was positively correlated with a longer time to successful roping (r = 0.76, *p* = 0.01) and recumbency (r = 0.94, *p* = 0.002), and worse (higher) induction score (r = 0.74, *p* = 0.03) and restraint score (r = 0.78, *p* = 0.01), but was not correlated to drug doses used.

Giraffes were approached to be cast down with ropes within 4.9 ± 1.3 min (range 3–6.6 min) when signs of adequate sedation were apparent, such as high gaited and accelerated ambling or galloping, star gazing, and no reaction to external stimuli. Recumbency was achieved 5.6 ± 1.4 min (range 3.5–7.6 min) after darting, and, in all the giraffes, it was reached in less than a minute from the beginning of the roping procedure. In all individuals except for two, the induction was rated as excellent (median induction score: 1, range 1–3), characterised by severely ataxic giraffes that were quickly roped down by the capture team, and safely reached recumbency. Naltrexone was administered in the jugular vein within 6.3 ± 1.4 min (range 5.1–8.2 min) and in less than a minute from when the individuals reached lateral recumbency. Five giraffes fell in the right lateral decubitus, whereas four giraffes fell in the left decubitus. None of them showed regurgitation at any stage. The duration of the recumbency phase under manual restraint was 13.3 ± 3 min (range 10.5–18 min). The median restraint score was 1 (range 1–3) and was rated excellent in six individuals out of nine. The immobilisations were safe except in one individual that required caution due to occasional but potent leg kicks. Most of the giraffes showed mild attempts to lift the neck at the beginning of the recumbency and occasionally moved the legs, but, within a few minutes, they accepted the manual restraint. There was no specific reaction to painful stimuli like arterial punctures. There were no correlations between differences in etorphine or azaperone doses in mg/kg and restraint score, which was also not correlated with induction time and score.

RR remained stable throughout the restraint (T1: 32.2 ± 7.3, T2: 31.8 ± 8.5, T3: 31.9 ± 8 breaths/minute), whereas HR slightly increased over time (T1: 68 ± 21, T2: 70 ± 16, T3: 73 ± 15 beats/minute). Rectal temperature ranged in different individuals from 37.0 to 38.5 °C (mean values in T1: 37.8 ± 0.4, in T2: 38.0 ± 0.5). NIBP was recorded at T2 only, and the mean values were 154 ± 22 mmHg for the systolic pressure, 122 ± 16 mmHg for the diastolic pressure, and 132 ± 16 mmHg for the mean arterial pressure (MAP). SpO_2_ values showed hypoxaemia in all individuals, which ranged from 72% to 80% (T1: 78 ± 3, T2 77 ± 1, T3: 77 ± 2). These values, however, did not correlate with SaO_2_ obtained from the blood gas analyses, as the difference between the two values in each individual was statistically significant (t = 3.6; *p* = 0.0018). There were no correlations between differences in etorphine or azaperone doses in mg/kg and HR, RR, T, SpO_2_, and NIBP. No significant differences were seen in these variables over time.

Arterial blood was drawn at T2 from seven individuals only, and the results of the arterial blood gas analysis are summarised in Table 7. The results from each individual are also shown in Table 8 with other variables such as RR, SpO_2_, and P(A-a) in order to better highlight the pathophysiological mechanism that occurred in the giraffes and the compensatory response. Furthermore, since negative correlation was demonstrated by the Spearman correlation test between excitement and HCO_3_^−^ (r = −0.83; *p* = 0.005), details on each individual’s induction-induced excitement score (1 = low, 4 = high) were also included in Table 8.

The BGA showed a slight to moderate metabolic acidosis in all individuals, characterised by bicarbonates variously decreased in comparison to ungulate reference values. In four individuals out of seven, PaCO_2_ was lower to what would normally be expected at this altitude [27]. Hypoxaemia, defined as PaO_2_ values lower than 66 mmHg, which is the expected PaO_2_ at the capture site altitude, was present in three individuals only, whereas the calculated SaO2 from the BGA were higher than 90% in four individuals out of seven. In all the individuals, the electrolytes were within ranges previously reported in giraffes [28]. The mean alveolar-to-arterial oxygen tension gradient P(A-a)O_2_ was 18.2 ± 5.4 mmHg, with values <20 mmHg in six individuals out of seven. As highlighted by the Spearman correlation test, individuals with higher induction-induced excitement scores had lower bicarbonates, whereas there were no statistical correlations for the other variables with the degree of excitement. No variables measured by BGA were correlated with the differences in etorphine or azaperone doses in mg/kg.

When the manual restraint was lifted, all the giraffes were able to stand up immediately, on average 18.8 ± 2.9 min from the darting. The median recovery score was 1 (range 1–2), with seven individuals rated excellent, and they stood up in a coordinated manner at the first attempt. The loading procedures on the chariot were rated good (median value 2, range 1–2), and it took 3.9 ± 2.4 min to safely load the giraffes. There were no correlations between differences in etorphine or azaperone doses in mg/kg and recovery and loading times and scores, which were not correlated to induction times and score, startle response, excitement, and restraint score.

The giraffes were transported in the chariot for 10-25 min and no complications occurred during the transport or the unloading procedure in the boma. During the 14 days in the boma, the nine giraffes appeared to be healthy and their vital functions were maintained. No complications such as re-narcotisation or signs of capture myopathy were observed during the follow-up period.

## 4. Discussion

This is the first study to evaluate an immobilisation protocol specifically in Masai giraffes. Since taxonomic classifications might be quickly changing, and Masai giraffes could emerge as a species itself, it is important to begin to have a species-specific approach from a veterinary point of view. Masai giraffes diverged 1.25 to 2 million years ago [1], so it would not be surprising to find differences, for example, in drug sensitivity as it is anecdotally reported from field veterinarians for Masai giraffes, and as is recognised for other similar species [1,7,29]. Etorphine-azaperone combination and its early reversal with naltrexone provided safe and reliable immobilisations and translocations in the nine free-ranging Masai giraffes. The doses used in this study provided quick and smooth inductions, which allowed all the individuals to reach recumbency safely in a challenging area characterised by thick vegetation and rough terrain. The purpose of the knock-down phase was to have an early ataxic state in order to cast the giraffes with ropes as soon as possible and lead them to reach recumbency in suitable spots to avoid overexertion on one side, and injuries on the other side [10].

All the individuals showed the first signs of drug effects between 1.5 and 3.4 min, and they were evaluated as sedated enough to be safely casted 3 to 6.6 min after they were darted. All the individuals fell recumbent in less than a minute with a minimum effort from the capture team, on average 5.6 ± 1.4 min after the dart injection. Seven out of nine inductions were rated as excellent, whereas only one individual posed a danger, as it was still galloping when approached by the capture team. Compared to other protocols used in ground-darted free-ranging giraffes, the mean time for the first signs of drug effects and time to recumbency in this study, were similar to those achieved with the protocol thiafentanil-medetomidine-ketamine [30], but quicker than in the protocols butorphanol-azaperone-medetomidine [31] and carfentanyl-xylazine [16] or quicker than in protocols used for captive giraffes, such as medetomidine-ketamine [32], etorphine-medetomidine-ketamine, thiafentanil-medetomidine-ketamine [30,32], and etorphine-acepromazine [33]. Times of etorphine-azaperone combination for the first signs of sedation and recumbency also resulted quicker than in combinations such as medetomidine-ketamine [11] or thiafentanil-medetomidine-ketamine [30] used in helicopter-darted free-ranging giraffes. However, the chase with the helicopter might have elicited a greater stress response that influenced not only the physiological balance of the individuals but also the drug effects.

Even though there were no correlations between the etorphine and azaperone doses and the times to first drug effects and induction, we have seen some individual variability in the rapidity and depth of the drug effects. A possible explanation is that, because the body weight was estimated, errors in the given dosage per kg might have occurred. The mg/kg doses are displayed only in order to make an easier comparison with other studies and must be taken with caution. Another possible explanation is that it has been demonstrated, in humans, that there is different genetic-mediated individual sensitivity to opioids. A common single nucleotide polymorphism, A118G, in the μ-opioid receptor gene (OPRM1), can affect opioid function and, consequently, has been suggested to contribute to individual variability in drug response to pain and drug addiction, through a regulation of the quantity and distribution of opioid receptors [34,35,36,37]. Although a similar allele and mechanism has been found in other species like mice and dogs [35,37], this field has not yet been investigated in wildlife, where it could provide an explanation for the many differences in drug response seen at individual and species’ level. In addition, the higher opioid-related excitement seen in some individuals, which was significantly correlated to longer induction times, might be caused by individual differences at the OPRM1 locus [35].

The doses of etorphine, azaperone, and naltrexone used in this study were based on previous experiences of the authors. The etorphine dose was higher compared to other opioid-based protocols described in giraffes [7,30,32,33], while azaperone was lower than previously reported [8,31]. The use of high doses of opioids for the knock down of capture myopathy-sensitive herbivores is common practice in order to obtain a fast recumbency and decrease the risk of overexertion and resulting homeostatic consequences such as acidosis and hypoxaemia [7,8]. Giraffes are particularly sensitive to both overexertion as well as to opioid side effects. For this reason, when capturing them for short field procedures, full antagonism is usually provided as soon as the giraffe reaches the ground [7,8,9,10]. They are then manually restrained, but this greatly limits what can be done to them and might also cause psychological stress with deleterious physiological consequences [8,10,12]. The addition of azaperone to the combination might add a few benefits. The synergism between etorphine and azaperone could have reduced induction time and improved immobilisation quality, thanks to the additional tranquilisation and anxiolytic effect provided, which could decrease the stress of the individuals and provide greater safety for the capture team [12,29,38]. Our results confirm that, at the doses used, a combination of 8–9 mg of etorphine and 50 mg of azaperone were adequate in knocking down subadults and young adult Masai giraffes quickly and with limited excitement, as shown by the limited homeostatic alterations that occurred and the safety of the restraint. In order to further decrease the opioid-related excitement shown in some individuals, future studies should evaluate multimodal drug protocols that include the addition of sedatives, since, by working in synergism with opioids, they might decrease induction times and excitement. Naltrexone is a long-acting pure μ antagonist used for the antagonism of opioid immobilisation in wildlife [17]. Despite naltrexone’s half-life being longer than that of etorphine, re-narcotisation has been observed between 2 and 72 h after immobilisation, even though the precise mechanism of re-narcotisation is not fully known [17,18]. In the early days, doses of naltrexone up to 100 mg/mg etorphine were used, while, more recently, between 5 and 25 mg naltrexone per milligram of etorphine have been reported to be effective without apparent re-narcotisation [7,8,17]. In a study performed in goats, the minimum effective naltrexone dose was 20 mg/mg etorphine, whereas re-narcotisation signs showed up between 20 and 133 min after antagonisation and lasted for 2 to 8 h when lower doses were given [17]. In our study, naltrexone at 3 mg/mg etorphine was administered intravenously as soon as the giraffes were recumbent. Even though it was challenging to record the exact time of recovery as it overlapped with the induction phase movements, the giraffes were clearly awake during the physical restraint. The mild hypoxaemia and the adequate cardiorespiratory function seen in the individuals also suggest that etorphine was reversed, as those would have likely been seriously compromised before naltrexone was given. However, the quick times and quality of recoveries and the 14 days boma follow-up are the proof that naltrexone at 3 mg/mg etorphine was effective in reversing etorphine in all the nine Masai giraffes. The small number of individuals in our study, and the fact that our results are innovative and in contradiction with previous reports in other species [17,39] suggest that further research with a larger sample size is needed, and that this dose of naltrexone should be used with caution until further work has been done on its use in giraffe to confirm its safety. However, novel research on polymorphism at OPMR1 could provide an explanation for species and individual variation in sensitivity not only to opioids, but also for their antagonists, i.e., by coding for different opioid receptor stability and brain expression, which can alter the binding of the antagonist [40]. The effect of naltrexone, which is used for treating alcoholism in human medicine, also seems to be moderated by the Asn40Asp single-nucleotide polymorphism of the μ-opioid receptor gene, and, moreover, to vary substantially as a function of ethnic background [41]. Although this is an emerging research topic and is currently under further development as it is being used in the novel field of human precision medicine, it is promising and could find a unique application for the advancement of wildlife immobilisation research.

After naltrexone was administered, the giraffes were manually restrained by the capture team for 10–18 min, which was the time that was needed to position the halters needed for the chariot loading. All the individuals were calm and occasionally made weak attempts to stand up at the beginning of the restraint, which generally decreased over time. To evaluate the restraint quality, an ad hoc descriptive score that considered the peculiarity of the protocol in which the knockdown drug was antagonised was created. The descriptive score scale differed from classic anaesthetic scales that consider the depth of anaesthesia and the presence of reflexes, but considered the overall giraffe fight reaction and safety working conditions for the capture team (Table 4). Six giraffes out of nine displayed an excellent level of restraint during which, even if they were awake, minimum fight reactions occurred and the safety conditions were appropriate, whereas only one giraffe required caution for occasional but potent kicks. Although no reactions to painful stimuli like arterial punctures were seen, this combination might not be appropriate for more painful procedures, as the early reversal with naltrexone also displace the analgesia provided by opioids. In some species, manual restraint has more controversial effects than the use of potent drugs for chemical restraint in the development of capture stress and its consequences such as catecholamine release, hyperthermia, reactive oxygen species (ROS) production, and cell damage [12,42]. Since, in our study, naltrexone was administered at the beginning of the immobilisation, giraffes were technically awake, and potentially more prone to psychological stress. The administration of tranquilisers and sedatives during capture procedures are recommended practices that appear to decrease the incidence of capture myopathy and improves survival rates [12]. The addition of azaperone might have provided additional tranquilisation and anxiolytic action that helped to keep the stress to a minimum during manual restraint, while not compromising the physiological compensation needed after the post-capture homeostatic derangement. A better restraint score was not correlated to etorphine, azaperone, or naltrexone doses, but was correlated to a lesser degree of excitement. We speculate that excitation-related poorer immobilisation could have been a consequence of catecholamines release by etorphine, which continued to display some excitation even when etorphine was reversed. However, since, in the same giraffes that had poorer restraint quality, their recovery and chariot loading were smooth, it showed that the low dose of naltrexone administered was efficient and was not correlated to the restraint quality level.

Few studies report physiological variables and blood gas analyses in detail in free-ranging restrained giraffes. Heart rate in our study slightly increased over time from 68 to 73 bpm, but remained within resting giraffe heart rates measured by telemetry or in trained unanaesthetised giraffes [43,44]. It was higher than in alpha2-agonist-based protocols, as these are commonly reported to cause bradycardia [45]. Etorphine stimulates catecholamines release, which, in many species, results in tachycardia [20,42], whereas, in the only study described in giraffes where etorphine was not immediately reversed, heart rate did not increase [33]. In our study, etorphine was immediately reversed with naltrexone, and, as such, the physiological variables reported in this study were technically collected under the effect of solely azaperone, which does not directly influence cardiac frequency [20]. The mean respiratory rate was 32 bpm and did not show variation over time. This value was lower than in other protocols described in immobilised giraffes with combinations such as medetomidine-ketamine [11] and butorphanol-azaperone-medetomidine [31], but was higher than in non-reversed opioid-based protocols such as etorphine-acepromazine [33] and etorphine or thiafentanil-medetomidine-ketamine [32]. It was also higher than the giraffe resting respiratory rate, which is 8–15 bpm depending on the age and size [13]. A slight increased respiratory rate is advocated in free-ranging giraffe opioid captures as it means that depression of the bulbar respiratory network does not occur and that a proper compensative response to balance the capture-related acidosis is maintained. In our study, since a low dose of naltrexone was administered, the presence of an adequate respiratory function is particularly important, as it highlights that the reversal dose was effective in the short term. A slightly increased respiratory rate, compared to giraffe resting rates, might have been beneficial in order to compensate metabolic acidosis, as demonstrated by the lowered PaCO_2_ levels, when compared to normal ranges and measured by the BGA. Furthermore, since the animals were technically awake, the absence of severe tachypnoea, which could be a consequence of manipulation stress, could be explained by the adjunct of azaperone in the dart, which seems to have been effective in tranquilising the giraffes.

Rectal temperature slightly increased throughout time, but it never exceeded 38.5 °C in any individuals. In a study conducted in impala, hyperthermia was not primary related to the effects of drugs such as metabolic and vascular effects of catecholamine release, environmental conditions, or physical activity but rather appeared to be strongly related to the level of initial stress (startle response) in response to capture [42]. Capture-induced hyperthermia may contribute to the development of capture myopathy [12,46]. Hence, its monitoring and prevention requires particular care, especially when using opioids that alter the thermoregulation mechanism [7]. In our study, the rise in rectal temperature was not correlated to startle response or excitement scores nor to longer times for induction. The small sample size might have limited our ability to detect this, but the fact that no severe hyperthermia was evident might find an explanation with the fact that the giraffes were captured in a highly touristic area where most animals are habituated to vehicles.

Due to the distance between their heart and their head, giraffes have developed a higher blood pressure compared to other mammals, and, when under anaesthesia, a minimal MAP of 120 mmHg is required in order to allow renal function [47]. In our study, in all individuals, the MAP recorded were above 120 mmHg, but since NIBP in giraffes is considered imprecise and inaccurate in estimating the true value, a further investigation deploying invasive blood pressure monitoring is needed [47]. Azaperone can reduce the hypertensive effects of the opioids thanks to its affinity to alpha1 receptors, which produces peripheral vasodilation and reduces mean arterial blood pressure. Since, in our study, etorphine and its side effects were immediately reversed with naltrexone, more research on the possible occurrence of hypotension in the awake giraffe would be needed [20,29].

Pulse-oximetry in our study showed moderate to severe hypoxemic values between 72% and 80%, which is similar to that reported with medetomidine-ketamine combination [11]. In accordance with the results reported by Bertelsen et al. [47], these values were statistically different from the values obtained from blood gas analyses, where calculated oxygen saturation was corrected according to the rectal temperature with values above 90% in four individuals. Furthermore, Bertelsen et al. [47] found the pulse-oximetry tended to overestimate haemoglobin oxygen saturation, but, in our data, the pulse-oximetry gave lower values when compared to SaO_2_. Accuracy of pulse oximeters and failure to produce a reading can vary widely between different models of pulse oximeters as well as between different species and skin pigmentation. Therefore, the readings should be interpreted with care [48]. Nonetheless, the values of SaO_2_ might not be accurate since they are calculated from an algorithm based on a human oxygen dissociation curve (ODC), which does not reflect the actual SaO_2_ in giraffes, as it is likely that the ODC is left shifted in this species. This is similar to other megaherbivores [49]. Although the calculated SaO_2_ values are not representative of the actual value of haemoglobin saturation, and, as such, cannot be used to validate the pulse-oximeter sensitivity, the fact that, in our study, the measured levels of PaO_2_ are mostly within physiologic ranges suggests that the SpO_2_ values recorded with the pulse-oximeter are not reliable.

Blood gas analyses revealed a slight to moderate metabolic acidosis with decreased values of bicarbonates in all individuals [27], whereas non-increased values of PaCO_2_ likely shows that there was no respiratory component for the acidosis. Although values of blood lactate were not available, it is likely that the acidosis was caused by an increase in lactic acid as commonly occurring in free-ranging herbivore capture. This is also supported by an overall increase in the anion gap [50]. Nonetheless, giraffes that had a greater excitement score, as well as those that, as a result, had longer times to casting and recumbency and have run for longer times or distances, had a significantly greater decrease in bicarbonates. The giraffes with worse excitement scores and lower bicarbonates did not show more severe acidosis (Table 8). Indeed, initial respiratory compensation occurred in some individuals seen as decreased PaCO_2_ values compared to reference values. PaO_2_ and SaO_2_ were also higher in these individuals whereas P(A-a)O_2_ was lower, which is likely a consequence of an increased respiratory rate within the compensatory response. The occurrence of respiratory compensation is particularly important in opioid-based protocols, as these depress the respiratory driven response to carbon dioxide, which leads to a potentially uncompensated acidosis and homeostatic catastrophe [7]. A similar or worse metabolic acidosis was reported in captive giraffe immobilised with medetomidine-ketamine-etorphine/thiafentanil and medetomidine-ketamine combinations, and, in a study of helicopter-darted free-ranging giraffes, it showed improvement within 30 min [11,32]. A limit of our study is that we took arterial samples only once, at the beginning of the restraint. Therefore, it is not possible to know if an improvement of acidosis occurred as well as in our individuals.

Considering the PaO_2_ that would be expected in a normal awake animal at the altitude of the capture area, only three giraffes could be considered mildly hypoxaemic with values of PaO_2_ lower than 66 mmHg. P(A-a)O_2_ was not elevated (normal <20 mmHg) except in one individual (27.4 mmHg). These results highlight that cardiorespiratory function was maintained after the early administration of naltrexone, and, also in those individuals with a mild hypoxaemia, there was no oxygen diffusion impairment or physiological right-to-left intrapulmonary shunting of blood. Even though in our study oxygen was not administered and minor hypoxaemia was recorded in few individuals only, hypoxemia is a common eventuality in giraffe immobilisations [10,11,30,32], thus oxygen insufflation should always be provided even in field situations to prevent poor tissue oxygenation [32,48]. In case of oxygen demand, giraffes have evolved a mechanism to increase the respiratory rate and oxygen diffusing capacity, rather than increasing the tidal volume due to anatomical constraints such as small lung volume and low lung compliance [13]. This inability to increase the tidal volume indicates the need for focussing attention on preventing acidosis and hypoxaemia in giraffes, as they might struggle more than other mammals to compensate a large homeostatic derangement [13]. Due to their long trachea, it is unclear whether an increased and shallow respiration would be able to overcome the dead space in the upper portion of the respiratory tract, as an increased respiratory rate would exacerbate the hypoxaemia [11]. Recent studies, however, have demonstrated that, since the trachea is narrow, giraffes have a low dead space-tidal volume ratio [51,52], but this might impose a high respiratory resistance during physical exercise, which is partially compensated by the longer inspiratory time and due to no pause between inspiration and expiration [13]. In our study, an increased respiratory rate accounted for an improvement of gas exchanges, as demonstrated by an increase in oxygenation and a decrease in carbon dioxide. This shows that the oxygen demand-driven mechanism was functionally maintained despite the drugs used.

Sodium and chloride were within free-ranging giraffes’ ranges, whereas potassium was similar to values reported in captive giraffes, but lower than in free-ranging captured giraffes [28]. Potassium elevations are due to high levels of capture stress since it results from muscle damage and is a major component in the pathogenesis of capture stress and myopathy [18]. Non-increased values of potassium recorded in this study supports that etorphine-azaperone combination kept the stress to a minimum, and, even though some excitement occurred, it had little consequences. Sodium in this study was lower than in other studies in free-ranging and captive giraffe [28], and the differences seen may be because of the small sample size and narrow range of values.

The excitement score evaluated the increased motor activity, which was likely due to both opioid excitation and anxiety. The startle response score was focused on the behavioural stress reaction of the individual when approached by the darting vehicle. In the giraffes of this study, the startle response score ranged between 1 and 3, and it did not involve any increased physical activity (i.e., full speed chase). The reaction after the dart impacted the animals was not taken into consideration as all the giraffes showed little reaction to it, such as a galloping for a few seconds, after which all the giraffes returned to their previous activity. Most capture events are naturally likely to induce an acute stress response, which can be defined as psychological stress [46]. This level of stress has higher influence on the development of hyperthermia and related homeostasis disturbance, than the physical activity itself or drug effects [42]. Although acute or chronic stress responses are usually evaluated respectively through the measurement of catecholamines or cortisol and metabolites, the effects of habituation can also be evident in the animal’s behaviour [42,53]. In our study, we used an ad hoc descriptive score based on behavioural reactions such as startle response to darting to define the presence of psychological stress of the giraffes. If alarmed, a giraffe can go quickly from a walk to a fast gallop of up to 56 km/hour and can sustain this for many kilometres [9]. In our study, the giraffes reacted differently to the sight of the car. A few stood still and others walked away to keep some distance. However, compared to helicopter captures among giraffes, the stress reaction displayed was minimal. Since we did not observe any correlation between higher startle response developed during the darting operation and alterations of the acid-base or cardiorespiratory function, it is likely that vehicle-darted captures in giraffes in highly touristic areas have a minimum impact on their stress response. Further investigations involving helicopter-based capture would be required in order to understand how a greater psychological stress influences the homeostatic balance and how the use of capture techniques that consider the species-specific susceptibility to different stressors can prevent it. In our study, the startle response during the darting operation and the drug-related excitement were not correlated. Regardless of the initial stress, our results show that the induction-induced excitement caused a higher degree of bicarbonate loss and, indirectly, metabolic acidosis. It is likely that it was due only to the increased physical activity resulting from the excitation, which likely results from the development of hypoxaemia and lactic acidosis. Psychological and physical stress are the two sides of the same question, which is capture stress. Our opinion is that it is fundamental to differentiate and deeply investigate their roles in the development of acid-base disturbance, as they have different prevention strategies.

The homeostatic derangements seen in this study resulting from a higher excitement were not severe. Usually, it will take time for the heart rate, blood pressure, and respiration to normalise once giraffes are restrained and reversed [9]. Thus, it is likely that the cardiorespiratory function we observed had further improved rapidly beyond our short-term monitoring. It is likely that the practice of roping the giraffes not only prevents injuries, but also decreases the physical activity during induction, which limits the overexertion effects. Recoveries in all the giraffes were safe and smooth. When the neck restraint was lifted, they immediately stood up in a coordinated manner at the first attempt between 16 and 27 min after the initial darting. The recovery score was excellent in all individuals, except for two that we rated as good. Diprenorphine is often preferred over naltrexone for giraffe translocations for its partial agonist/antagonist effect, even though it has a greater re-narcotisation risk [17]. Pure antagonists such as naltrexone induce complete reversal of the opioid immobilisation and, as such, giraffes have been considered too excited to be safely managed and loaded. In our study, the combination of naltrexone, which fully reversed etorphine even if administered at low doses, and the addition of the tranquiliser action provided by azaperone allowed coordinated recoveries and loading procedures, as the giraffes were conscious but calm. The use of sedatives or tranquilisers are usually not advised for giraffe translocations, as they are reported to decrease the coordination and balance [7,9]. In our study, all the giraffes regained coordination from the beginning and maintained it during the loading procedure and during the 10-25 min drive to the boma. Azaperone is not reversible, but it did not prevent satisfactory recoveries, as giraffes were loaded in the chariot quickly and effectively in less than 4 min, and no signs of excessive tranquilisation were evident. Tranquilisation has likely made the transport to the boma smoother and uneventful, and it might have reduced the giraffe stress and their fight reactions that can be deleterious during the loading procedures. Recovery and loading times as well as scores were not correlated to drug doses nor to alterations seen in the cardiorespiratory function and blood gases, which confirms that no serious physiological compromise occurred from the capture.

The boma period represented a valuable tool that allowed to closely monitor the giraffes and evaluate possible tardive complications. During the 14 days of a follow-up, particular attention was taken toward signs of re-sedation and capture myopathy. Since re-narcotisations have been reported to mostly occur within the first hours and lasts 2–8 h, our monitoring protocol was likely to be adequate in spotting possible signs of re-narcotisation such as fear loss, aimless wandering, ataxia, and depression [17]. The giraffes maintained their vital functions and remained healthy during the monitoring period.

## 5. Conclusions

The results in this preliminary study showed that the etorphine-azaperone combination provided reliable immobilisations in ground darted Masai giraffe with adequate physiological and handling safety. Early reversal with a low naltrexone dose (3 mg/mg etorphine) was successfully obtained and the two weeks boma follow-up excluded the occurrence of re-narcotisation or capture myopathy. Although a control group was not available, the addition of azaperone provided tranquilisation that likely increased the restraint quality and facilitated a smooth loading and safe chariot transport. Casting giraffes with ropes is a valuable tool to minimise the overexertion, especially when due to the combination of high opioid doses and tranquilisers, the giraffes move with a slow high stepping gate, which minimises the injury risks for the capture team. Startle reactions and induction-induced excitement both occur during the capture of giraffes, and, since they have different pathways and origins, it is important to analyse them separately whenever possible. Although this is only a pilot study on a small number of individuals, the encouraging results obtained can provide an initial insight into the causes of the stress induced in the giraffe. Opioid-related excitement accounted for greater physiological derangement, and further research on the use of multimodal anaesthetic drug mixture might provide a solution to mitigate it.

The use of ad hoc descriptive scores seems to be a useful tool for evaluating the predisposing factors in influencing the morbidity in chemical and physical restraint, and might give important information if used in a larger study. Systematic monitoring that includes the analysis of arterial blood gases should always be adopted to be able to detect physiological changes that might otherwise go unnoticed. A good knowledge of the physiological responses of the animals is important to ensure the success of a capture operation by preventing possible complications.

## Figures and Tables

**Table 1 animals-10-00322-t001:** Startle response score.

Startle Response Score	Description
1	No reaction when approached by the darting vehicle, keep on with the previous activity or stop to observe the vehicle
2	Suspicion over the darting vehicle. Intermittent walking away for maximum 2 min only when the vehicle is close
3	Keep distance from the darting vehicle, walking away for up to 5 min before being successfully darted
4	Sustained chasing involving a high velocity gallop before being successfully darted

**Table 2 animals-10-00322-t002:** Induction-induced excitement score.

Excitement Score	Description
1	Excitement shorter than 1 min, mainly slow, high gaited step ambling
2	Excitement phase of maximum 2 min, involving low velocity ambling or galloping
3	Excitement phase between 2 and 4 min, including a high velocity gallop
4	Sustained high velocity gallop, characterised by unsafe and repetitive unsuccessful roping attempts

**Table 3 animals-10-00322-t003:** Induction score.

Induction Score	Description
1	Excellent induction; motionless star gazing or slow ataxic gait and no reaction to the capture team’s approach. When roped, the recumbency is quickly achieved without struggling. Minimum risk for the safety of the giraffe and the capture team.
2	Good induction. The individual shows a severe ataxia and an accelerated high gait ambling when approached by the capture team. When roped, shows no fight response and reaches recumbency easily.
3	Fair induction. The individual shows a severe ataxia but gallops. While being entangled by the capture team’s ropes, it is still strong and keeps on galloping for meters before falling.
4	Poor Induction. The giraffe shows a severe excitement and has been galloping for minutes. Failure of roping attempts due to a fight reaction. Dangerous to rope, it would require a second dosing.

**Table 4 animals-10-00322-t004:** Restraint score.

Restraint Score	Description
1	No attempts to stand or fight. The giraffe is alert but quiet. Safe handling.
2	Weak attempts to stand. The giraffe is quiet, but manual restraint is essential. Safe handling.
3	Repetitive attempts to stand. The giraffe is strong and sometimes kicks. Caution required.
4	Strong attempts to stand, excitation, and aggressive behaviour. Extremely dangerous situation.

**Table 5 animals-10-00322-t005:** Recovery score.

Recovery Score	Description
1	Excellent recovery. Standing at the first attempt as soon as manual restraint is lifted, immediate balance, and coordination.
2	Good recovery. Standing at the first attempt once manual restraint is lifted and balance is gained after a few steps.
3	Fair recovery. Few attempts before standing. Weakness and poor balance in the first several seconds. Risk of injuries.
4	Poor recovery. Struggle to stand with one or more attempts and poor balance once standing. High risk of injuries.

**Table 6 animals-10-00322-t006:** Loading score.

Loading Score	Description
1	The giraffe has good coordination and easily follows the rope guidance to load into the chariot at the first attempt.
2	The giraffe has a good balance but is not loaded at the first attempt as it does not follow correctly the rope guidance. The giraffe is calm and the situation is not dangerous.
3	The giraffe has a poor balance and falls during the loading procedure. Repetitive attempts before a successful loading. Unsafe procedure.
4	The giraffe has a good balance, but makes attempt to fight the ropes, i.e., kicking or running away. Loading is difficult and dangerous for the team and the giraffe.

**Table 7 animals-10-00322-t007:** Arterial blood gas analyses analysed from samples collected at T2.

Variable	Mean	Standard Deviation
pH	7.23	0.05
HCO_3^−^_ (mmol/l)	12.9	1.2
PaCO_2_ (mmHg)	34	4
Anion Gap (mmol/l)	28	3
tCO_2_ (mmol/l)	13.8	1.3
BE (mmol/l)	−12.4	1.8
PaO_2_ (mmHg)	67	8
tHb (g/dl)	12.5	1.1
SO_2_ (%)	87	5
Na (mmol/l)	142	1.2
K (mmol/l)	4.6	0.8
Cl (mmol/l)	105	4.6

**Table 8 animals-10-00322-t008:** Most relevant BGA values and physiological variables organised according to the individual’s induction-induced excitement score.

Excitement Score	HCO_3−_	pH	PaCO_2_	BE	RR	PaO2	AG	Cl_−_	SaO_2_	SpO_2_	P(A-a) O_2_	PAO_2_	Pb
1	14.5	7.2	41	−12.2	32	58	25.9	108	81	77	19.2	77.2	609.8
1	13.5	7.22	36	−12.2	30	55	26.7	108	80	78	27.4	82.4	611.1
2	13.4	7.26	32	−11.4	32	67	26.1	108	91	80	19.7	86.6	612.1
2	13.3	7.28	31	−10.7	32	68	33.4	97	91	79	18.9	86.9	608.4
3	12.7	7.26	31	−11.6	50	78	27.9	109	92	74	9.5	87.6	611.7
3	11.9	7.25	29	−12.5	36	75	28.5	109	92	80	14.6	89.5	611.6
4	10.8	7.14	35	−16.2	32	66	33.7	102	84	84	17.6	83.6	612.1

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
