# Peer review of "Etorphine-Azaperone Immobilisation for Translocation of Free-Ranging Masai Giraffes (Giraffa Camelopardalis Tippelskirchi): A Pilot Study"

_animals, 2020, doi:10.3390/ani10020322_

Round 1

Reviewer 1 Report

None of our suggestions have been taken into consideration.
It is recommended to improve the discussions
inserting what previously indicated.

In discussions, you need to add comparison notes with
the recent bibliography:

- possible influence of the
ambient temperature during anesthesia
- use of hyaluronidase
in the anesthetic mixture.

Reviewer 2 Report

The major issues have been addressed and the manuscript has improved, which has now allowed for a more detailed review.

General comments:

I still believe that this manuscript can be shortened and made more succinct, especially the discussion. There is still a section that is completely over-interpreted that needs to be removed, as it is conjecture. There are a number of grammatical errors that need to be corrected and I would strongly suggest that the whole manuscript be carefully reviewed for editorial soundness. In the attached document I have highlighted specific changes and where improvements can be made.

Author Response

This manuscript is a resubmission of an earlier submission. The following is a list of the peer review reports and author responses from that submission.

Round 1

Reviewer 1 Report

Although this is an interesting study and adds some information to the field of giraffe immobilization the findings are sometimes miss- and over-interpreted. Based on the fact that this data is preliminary I would strongly recommend that this manuscript be shortened to only report on key findings rather than make conclusions from data that is insufficient or over-analysed. The discussion section should be substantially shortened and revamped as this is where the majority of the over-interpretation occurs. There are also a number of terms that are used in correctly. 

Specific major comments: 

The authors should discuss and compare this data only to similar giraffe capture procedures as these animals were habituated to vehicles and could be darted from the ground. Animals that have to be darted from a helicopter will have a greater startle response and the effects of the drugs and capture combined will be very different. This comment specifically applies to line 371-375, but should be considered throughout the manuscript.

I don't believe that it is correct to use the terms psychological and physical stress and then try and separate events based on these terms i.e. physical stress for opioid-related excitation. Firstly it is pure speculation that these forms of stress are related to each event, and secondly it is likely that psychological and physical stress occurs in all the phases of this capture. I think it will be more appropriate to refer to these events for what they are i.e. a "startle response" from the darting procedure and "induction-induced excitation". Furthermore, equating all of the excitement in the induction phase to the opioids may not be correct, especially as there is likely to still be a large component of the psychological stress response that also drives this excitement.  

A major flaw to this study is that body weight of these animals was estimated, not measured. Either a validated method for this estimation should be provided, or all of the information about drug doses (mg/kg), the analysis done on these doses, and these results, should be removed from the manuscript.  

Additional data related to the stats should be included, not just p values i.e. r values for the correlations and t values for the T-tests.

After the naltrexone was administered it is no longer appropriate to refer to this period as immobilization as the effects of etorphine, the immobilizing drug, were antagonized. Possibly refer to this period as “manual restraint” as these animals were conscious but physically restrained.

I believe that the interpretation of the blood gases is wrong. Firstly it is not mentioned at which altitude the animals were worked on. Looking at the data it was unlikely to be at see level. Therefore the authors should consider calculating what the expect PaO2 of the animals would be using the alveoli gas equation and the predicted normal A-a gradient (10-20 mmHg). I suspect that they may find that their animals had PaO2 values that would be close to those expect at the altitude that they were working at i.e. that these animal may not all have been hypoxaemic but just had normally low PaO2 values for that altitude. Furthermore, the PaCO2 in most of the animals was below 35mmHg indicating that the animals were mostly hyperventilating (possibly from the stress of been consciously restrained) and therefore it is unlikely that they did not get “enough oxygen into their lungs (decrease intake of oxygen)” from hypoventilation (review line 544 specifically but also this whole paragraph). It may be useful to add the PAO2 and expected PaO2 values to table 8.

It is important for the authors to report whether the SaO2 was measured or calculated from the blood gas analyzer. If calculated it is likely that this calculation is from a human oxygen dissociation curve (ODC) and therefore may not reflect what the giraffes actual SaO2 were as giraffe are likely to have an ODC curve that is left shifted (i.e. lower p50 compared to humans). This limitation should also be discussed based on the comparison with the SpO2.

I’m concerned that no additional naltrexone dose was administered to these animals. A dose of 3mg per mg etorphine is extremely low and I believe that it poses a great risk for renarcotization. Therefore, I propose that the authors should mention that this dose should be used with caution until further work has been done on its use in giraffe, to confirm its safety.

I’m struggled to follow the discussion around respiratory rates (lines 488-497), especially that related to opioid-induced respiratory depression, as the data presented is after naltrexone was administered, so etorphine should not have had an effect around this time. I believe that the statement that azaperone inhibits opioid-induced respiratory depression is speculative, this has never been proven, but I may be wrong, but need to be convinced by good evidence i.e. a better reference needs to be used.

Another statement that is speculative is that potent opioids cause hyperthermia. Opioid drugs do alter thermoregulation, but some actually cause hypothermia. Generally the hyperthermia caused during capture is mostly stress-induced!

As mentioned above the discussion on pulse oximetery will need to be revised taking into consideration the limitations of both the pulse oximeter and blood gas device, in terms of the different ODC of giraffe.

Jargon terms should be avoided i.e. line 78 and 398 opioid toxicity is an incorrect term used for opioid-induced side-effects (toxicity causes direct toxic damage to tissues),  line 148 "lift" the opioid-related..., rather antagonize the opioid-related...... 

Line 95 the more important adverse effect would be hypotentsion (obviously from vasodilation)!

Line 189 provide manufacturer details of the digital thermometer.

Line 200 what type of probe was used, a clip (transmission) or reflectance probe?

Line 215 water “vapor” in the alveoli

Line 229 Table 5 what does “spills” mean?

Line 315 Use the word variable rather than parameter. Parameters are things that influence the variables that are measured.

Line 383 sensibility - sensitivity?

Line 395 reword sentence.

Author Response

Point 1: Although this is an interesting study and adds some information to the field of giraffe immobilization the findings are sometimes miss- and over-interpreted. Based on the fact that this data is preliminary I would strongly recommend that this manuscript be shortened to only report on key findings rather than make conclusions from data that is insufficient or over-analysed. The discussion section should be substantially shortened and revamped as this is where the majority of the over-interpretation occurs. There are also a number of terms that are used in correctly. 

Response 1: Thank you for your overall opinion and recommendation. We have revised the draft based on your suggestions.

Specific major comments: 

Point 2: The authors should discuss and compare this data only to similar giraffe capture procedures as these animals were habituated to vehicles and could be darted from the ground. Animals that have to be darted from a helicopter will have a greater startle response and the effects of the drugs and capture combined will be very different. This comment specifically applies to line 371-375, but should be considered throughout the manuscript.

Response 2: Thank you for this comment. We have highlighted in the discussion when the comparison was made with giraffes darted from a helicopter in order to consider the different stress these animals underwent to. However, we did not feel to exclude the data reported in those studies, as the knowledge on giraffe immobilization is such limited (especially reports on BGA and physiological variables) that it is important to mention them in our study, if the different capture conditions are reported and discussed. We hope that now the paragraph is more clear.

Point 3: I don't believe that it is correct to use the terms psychological and physical stress and then try and separate events based on these terms i.e. physical stress for opioid-related excitation. Firstly it is pure speculation that these forms of stress are related to each event, and secondly it is likely that psychological and physical stress occurs in all the phases of this capture. I think it will be more appropriate to refer to these events for what they are i.e. a "startle response" from the darting procedure and "induction-induced excitation". Furthermore, equating all of the excitement in the induction phase to the opioids may not be correct, especially as there is likely to still be a large component of the psychological stress response that also drives this excitement.  

Response 3: Thank you for your comment on this. Although the “startle response” in our study did not included any physical stress as there was no reaction such as running, the “induction-induced excitation” could have a component of psychological stress that we can’t exclude. We have modified the terms psychological and physical stress with the ones you have suggested. We believe that it is important tough to separate the events, and analyse them separately in the statistical analysis, as they have different consequences and in different individuals their presence can vary. I.e. in some giraffes we have observed that the startle response is huge (in giraffes living in remote areas or in helicopter darted giraffe) and have deleterious consequences for their homeostatic imbalance, while the induction-induced excitement in the same individuals can vary. In other giraffes, startle response is very little but they can have a great excitement after the first signs of sedations. In our opinion the two mechanism must be distinguished as they don’t have the same trend, and for example the induction-induced excitement can be modulated by the use of different drug cocktails, while startle response can’t, supporting the opinion that the effects of each “stress mechanism” must be evaluated.

Point 4: A major flaw to this study is that body weight of these animals was estimated, not measured. Either a validated method for this estimation should be provided, or all of the information about drug doses (mg/kg), the analysis done on these doses, and these results, should be removed from the manuscript.  

Response 4: Thank you for this observation. We have included in the text and in the bibliography the validated method used to estimate the weight of the giraffes.

Point 5: Additional data related to the stats should be included, not just p values i.e. r values for the correlations and t values for the T-tests.

Response 5: We have included data relative to statistics, such as t and r.

Point 6: After the naltrexone was administered it is no longer appropriate to refer to this period as immobilization as the effects of etorphine, the immobilizing drug, were antagonized. Possibly refer to this period as “manual restraint” as these animals were conscious but physically restrained.

Response 6: Thank you for this observation, we have edited the word “immobilization” in the text with “manual restraint”.

Point 7: I believe that the interpretation of the blood gases is wrong. Firstly it is not mentioned at which altitude the animals were worked on. Looking at the data it was unlikely to be at see level. Therefore the authors should consider calculating what the expect PaO2 of the animals would be using the alveoli gas equation and the predicted normal A-a gradient (10-20 mmHg). I suspect that they may find that their animals had PaO2 values that would be close to those expect at the altitude that they were working at i.e. that these animal may not all have been hypoxaemic but just had normally low PaO2 values for that altitude. Furthermore, the PaCO2 in most of the animals was below 35mmHg indicating that the animals were mostly hyperventilating (possibly from the stress of been consciously restrained) and therefore it is unlikely that they did not get “enough oxygen into their lungs (decrease intake of oxygen)” from hypoventilation (review line 544 specifically but also this whole paragraph). It may be useful to add the PAO2 and expected PaO2 values to table 8.

Response 7: Thank you for this suggestions, and we agree that it is important to add more information and describe the data presented with more precision. We have included the altitude in the text (material and methods).

The blood gas analyser calculates the barometric pressure, which value has already been used to calculate the alveolar pressure (PAO2) based on the measured barometric pressure. We have used a respiratory quotient =1, as it is the value generally used for ruminants (see Zeiler at al., 2017, reference n. 24 in the text). We have modified the table 8 as we included the value of barometric pressure for each giraffe (if you believe it is unappropriate, we can remove the barometric pressures), and have also included the expected and the PAO2.

By calculating the expected PaO2 according to the altitude where the capture took place, it results that only three animals had a mild hypoxaemia. We have included these data in the manuscript, and are glad to have received this comment as the new editing are interesting and more precise.

Point 8: It is important for the authors to report whether the SaO2 was measured or calculated from the blood gas analyzer. If calculated it is likely that this calculation is from a human oxygen dissociation curve (ODC) and therefore may not reflect what the giraffes actual SaO2 were as giraffe are likely to have an ODC curve that is left shifted (i.e. lower p50 compared to humans). This limitation should also be discussed based on the comparison with the SpO2.

Response 8: Thank you for this observation, the SaO2 was calculated from the blood gas analyzer and it is true that the algorithm is not specie-specific from giraffes. We have implemented the discussion including this limitation.

Point 9: I’m concerned that no additional naltrexone dose was administered to these animals. A dose of 3mg per mg etorphine is extremely low and I believe that it poses a great risk for renarcotization. Therefore, I propose that the authors should mention that this dose should be used with caution until further work has been done on its use in giraffe, to confirm its safety.

Response 9: We have added a comment on this in the manuscript.

Point 10: I’m struggled to follow the discussion around respiratory rates (lines 488-497), especially that related to opioid-induced respiratory depression, as the data presented is after naltrexone was administered, so etorphine should not have had an effect around this time. I believe that the statement that azaperone inhibits opioid-induced respiratory depression is speculative, this has never been proven, but I may be wrong, but need to be convinced by good evidence i.e. a better reference needs to be used.

Response 10: Thank you for this comment, we agree that the explanation around our interpretation of the respiratory compensation was not clear. We have elucidated in the text the explanation over the fact that the presence of increased respiratory rate is particularly important since naltrexone is given at a low dose, and it is essential to show that this dose of antagonist was able to reverse etorphine respiratory depression. We have also removed the statement over azaperone that might inhibit opioid-induced respiratory depression, as it has not been demonstrated in any peer-reviewed article.

Point 11: Another statement that is speculative is that potent opioids cause hyperthermia. Opioid drugs do alter thermoregulation, but some actually cause hypothermia. Generally the hyperthermia caused during capture is mostly stress-induced!

Response 11: We completely agree that hyperthermia during capture is mostly stress-induced, and have written different statements on this along the manuscript, citing proper references. None the less we agree that this sentence is ambiguous, so we have modified it to the following statement “Opioid related respiratory depression and alteration in thermoregulatory response commonly occur and worsen the oxygen debt and acid-base derangement and in case of stress-driven hyperthermia, further increase the metabolic demand [8].”

Point 12: As mentioned above the discussion on pulse oximetery will need to be revised taking into consideration the limitations of both the pulse oximeter and blood gas device, in terms of the different ODC of giraffe.

Response 12: We have expanded the discussion here as well.

Point 13: Jargon terms should be avoided i.e. line 78 and 398 opioid toxicity is an incorrect term used for opioid-induced side-effects (toxicity causes direct toxic damage to tissues),  line 148 "lift" the opioid-related..., rather antagonize the opioid-related...... 

Response 13: Thank you for these specific corrections and suggestions, we have edited the above mentioned and the last following comments, in the manuscript text.

Point 14: Line 95 the more important adverse effect would be hypotentsion (obviously from vasodilation)!

Response 14: Edited in the text.

Point 15: Line 189 provide manufacturer details of the digital thermometer.

Response 15: Included in the text.

Point 16: Line 200 what type of probe was used, a clip (transmission) or reflectance probe?

Response 16: Included in the text.

Point 17: Line 215 water “vapor” in the alveoli

Response 17: Edited in the text.

Point 18: Line 229 Table 5 what does “spills” mean?

Response 18: Edited in the text.

Point 19: Line 315 Use the word variable rather than parameter. Parameters are things that influence the variables that are measured.

Response 19: Edited in the text.

Point 20: Line 383 sensibility - sensitivity?

Response 20: Edited in the text.

Point 21: Line 395 reword sentence.

Response 21: Edited the sentence in the text.

Reviewer 2 Report

Altogether the manuscript deserves to be published. Some modifications could improve scientific interest.

The manuscript should read it an English-speaking man to correct some minor inaccuracies. 

In discussions, notes should be added on the possible influence of ambient temperature during anesthesia. Recommended bibliography:

Influence of Ambient Temperature and Confinement on the Chemical Immobilization of Fallow Deer (Dama dama). 2017. JOURNAL OF WILDLIFE DISEASES. - Costa et Al.

Reply to Arnemo and Kreeger: Commentary on Influence of Ambient Temperature and Confinement on the Chemical Immobilization of Fallow 2017. JOURNAL OF WILDLIFE DISEASES. - Costa et Al.

It is also advisable to mention the use of hyaluronidase in the anesthetic mixture. Recommended bibliography:

Hyaluronidase, with xylazine and ketamine, reducing immobilization time in wild cattle (Bos Taurus). 2019. LARGE ANIMALS REVIEW - Spadola et Al.

kind regards

Author Response

Point 1: Altogether the manuscript deserves to be published. Some modifications could improve scientific interest.

The manuscript should read it an English-speaking man to correct some minor inaccuracies. 

Response 1: Thank you for your suggestions, one of the co-authors, Prof. Richard Preziosi, is native speaking in English and has reviewed and corrected this last version of the manuscript.

Point 2: In discussions, notes should be added on the possible influence of ambient temperature during anesthesia. Recommended bibliography:

Influence of Ambient Temperature and Confinement on the Chemical Immobilization of Fallow Deer (Dama dama). 2017. JOURNAL OF WILDLIFE DISEASES. - Costa et Al.

Reply to Arnemo and Kreeger: Commentary on Influence of Ambient Temperature and Confinement on the Chemical Immobilization of Fallow 2017. JOURNAL OF WILDLIFE DISEASES. - Costa et Al.

Response 2: We thank you for this observation. In our study, environmental temperature was not excessive neither too low, and was very stable between different individuals. Indeed, we believe that the absence of hyperthermia that we experienced were mostly due to limited stress experience by the giraffes, mainly due to habituation to vehicles (highly touristic area) and not related to ambient temperature. We have cited in the text some references that support the theory that temperature is influenced mostly by stress and not much by the environmental temperature, and would be contradictory to add references saying the contrary.

Point 3: It is also advisable to mention the use of hyaluronidase in the anesthetic mixture. Recommended bibliography:

Hyaluronidase, with xylazine and ketamine, reducing immobilization time in wild cattle (Bos Taurus). 2019. LARGE ANIMALS REVIEW - Spadola et Al.

Response 3: Thank you for this observation. Although we believe that the use of hyaluronidase can be very helpful in many situations, in our study we did not use hyaluronidase.